# Repurposing a Lipid-Lowering Agent to Inhibit TNBC Growth Through Cell Cycle Arrest

**DOI:** 10.3390/cimb47080622

**Published:** 2025-08-05

**Authors:** Yi-Chiang Hsu, Kuan-Ting Lee, Sung-Nan Pei, Kun-Ming Rau, Tai-Hsin Tsai

**Affiliations:** 1School of Medicine, I-Shou University, Kaohsiung 824, Taiwan; jenway74@isu.edu.tw; 2Graduate Institutes of Medicine, College of Medicine, Kaohsiung Medical University, Kaohsiung 807, Taiwan; ayta860404@gmail.com; 3Department of Hematology Oncology, E-Da Cancer Hospital, I-Shou University, Kaohsiung 824, Taiwan; sungnanpei@gmail.com (S.-N.P.); liu07822@ms57.hinet.net (K.-M.R.); 4Division of Neurosurgery, Department of Surgery, Kaohsiung Medical University Hospital, Kaohsiung 807, Taiwan; 5Department of Surgery, School of Medicine, College of Medicine, Kaohsiung Medical University, Kaohsiung 807, Taiwan; 6Department of Surgery, Kaohsiung Municipal Siaogang Hospital, Kaohsiung 812, Taiwan

**Keywords:** simvastatin, triple negative breast carcinoma (TNBC), CDK4, cell cycle arrest, G1 phase, hyperlipidemia, repurposed drugs, breast cancer therapy

## Abstract

Triple-negative breast cancer (TNBC) is a highly aggressive and therapeutically challenging subtype of breast cancer due to its lack of estrogen receptors, progesterone receptors, and HER2 (Human epidermal growth factor receptor 2) expression, which severely limits available treatment options. Recently, Simvastatin—a widely used HMG-CoA (3-hydroxy-3-methylglutaryl-coenzyme A) reductase inhibitor for hyperlipidemia—has garnered interest for its potential anticancer effects. This study investigates the therapeutic potential of Simvastatin in triple-negative breast cancer (TNBC). The results demonstrate that Simvastatin significantly inhibits the proliferation of TNBC cells, particularly MDA-MB-231, in a dose- and time-dependent manner. Mechanistically, Simvastatin primarily induces G1 phase cell cycle arrest to exert its antiproliferative effects, with no significant evidence of apoptosis or necrosis. These findings support the potential repositioning of Simvastatin as a therapeutic agent to suppress TNBC cell growth. Further analysis shows that Simvastatin downregulates cyclin-dependent kinase 4 (CDK4), a key regulator of the G1/S cell cycle transition and a known marker of poor prognosis in breast cancer. These findings highlight a novel, apoptosis-independent mechanism of Simvastatin’s anticancer action in TNBC. Importantly, given that many breast cancer patients also suffer from hyperlipidemia, Simvastatin offers dual therapeutic benefits—managing both lipid metabolism and tumor cell proliferation. Thus, Simvastatin holds promise as an adjunctive therapy in the treatment of TNBC and warrants further clinical investigation.

## 1. Introduction

Triple negative breast carcinoma (TNBC) represents approximately 10–20% of all breast cancers and is characterized by the absence of estrogen receptor (ER), progesterone receptor (PR), and HER2 expression [1,2]. This lack of receptor targets significantly limits the effectiveness of current endocrine and HER2-targeted therapies, which are widely used in other breast cancer subtypes [2]. As a result, systemic chemotherapy remains the mainstay treatment for TNBC. However, TNBC is often more aggressive, exhibits higher rates of recurrence and metastasis, and is associated with poorer overall prognosis compared to other subtypes [3]. Moreover, the heterogeneity within TNBC poses an additional challenge, as it encompasses diverse molecular profiles with varying responses to treatment [4]. Despite advances in immunotherapy and PARP inhibitors for specific TNBC subgroups, there remains a critical need for novel therapeutic strategies that are both effective and accessible. This therapeutic gap underscores the urgency to explore alternative treatment options, including drug repurposing approaches, such as using statins with known safety profiles and emerging anticancer properties [5].

Simvastatin, a lipid-lowering agent commonly prescribed for hypercholesterolemia, functions by inhibiting 3-hydroxy-3-methylglutaryl-coenzyme A (HMG-CoA) reductase, the rate-limiting enzyme in the mevalonate pathway [6]. The mevalonate pathway is integral not only to cholesterol biosynthesis but also to the prenylation of key signaling proteins that regulate cell proliferation and survival. This pathway produces isoprenoid intermediates such as farnesyl pyrophosphate (FPP) and geranylgeranyl pyrophosphate (GGPP), which are essential for the post-translational modification known as prenylation. Prenylation facilitates the attachment of proteins like Ras and Rho GTPases to cell membranes, enabling their proper function in signaling pathways that control cell growth and survival [7]. In cancer cells, the mevalonate pathway is often upregulated, leading to increased prenylation of oncogenic proteins and promoting tumor progression. Inhibiting this pathway can disrupt the prenylation process, thereby impairing the function of proteins critical for cancer cell proliferation and survival [8]. Therefore, targeting the mevalonate pathway offers a promising strategy for cancer therapy by affecting both cholesterol synthesis and the prenylation of proteins involved in oncogenic signaling.

Preclinical studies have demonstrated that Simvastatin exerts multifaceted anti-tumor effects across a variety of cancers, including breast cancer, by inducing apoptosis, suppressing cell proliferation, inhibiting angiogenesis, and reducing metastatic spread [9]. These effects are believed to be mediated through interference with downstream products of the mevalonate pathway, such as farnesyl pyrophosphate (FPP) and geranylgeranyl pyrophosphate (GGPP), which are essential for the post-translational modification of oncogenic proteins like Ras and Rho [9]. In breast cancer models, Simvastatin has been shown to impair tumor cell viability, modulate cell cycle regulators, and sensitize cancer cells to conventional chemotherapeutics [10]. Notably, Simvastatin’s impact on triple-negative breast cancer, a subtype with limited treatment options, has garnered particular interest due to its potential to modulate key signaling pathways, including PI3K/Akt and MAPK [10] and to influence epithelial-to-mesenchymal transition (EMT) [11], a critical process in metastasis [12]. Clinically, retrospective cohort studies and epidemiological analyses have suggested a possible association between statin use and reduced cancer incidence or improved survival in breast cancer patients [13]. However, prospective clinical trials specifically evaluating Simvastatin as an anti-cancer agent remain limited. Ongoing research is necessary to clarify the optimal dosing, treatment timing, and potential synergistic effects with existing chemotherapies or targeted therapies [14]. Given its established safety profile, affordability, and widespread availability, Simvastatin represents a promising candidate for drug repurposing in oncology. Further elucidation of its molecular mechanisms and validation in clinical settings could pave the way for incorporating statins like Simvastatin into integrative treatment strategies for breast cancer and potentially other malignancies.

This study explores the potential anticancer effects of Simvastatin in TNBC, with a particular emphasis on its role in regulating cell cycle progression via CDK4. TNBC is a highly aggressive subtype of breast cancer that lacks hormone receptors and HER2 expression, leaving patients with limited targeted treatment options. Simvastatin, a commonly used lipid-lowering agent and HMG-CoA reductase inhibitor, has recently garnered attention for its potential anticancer properties beyond its cardiovascular benefits. The primary objective of this study is to evaluate whether Simvastatin can suppress the proliferation of TNBC cells, particularly MDA-MB-231, by inhibiting CDK4 activity and inducing G1 phase cell cycle arrest. By elucidating this cell cycle–related mechanism, we aim to provide preclinical evidence supporting the repositioning of Simvastatin as a potential therapeutic agent for TNBC.

## 2. Materials and Methods

### 2.1. Materials

Simvastatin, MTT [3-(4,5-dimethylthiazol-2-yl)-2,5-diphenyltetrazolium bromide], and dimethyl sulfoxide (DMSO) were purchased from Sigma-Aldrich (St. Louis, MO, USA). Simvastatin was dissolved in DMSO to prepare a stock solution (10 mM) and stored at –20 °C, protected from light. Working concentrations were freshly prepared by diluting the stock in complete culture medium immediately before use. Dulbecco’s Modified Eagle Medium (DMEM), fetal bovine serum (FBS), penicillin-streptomycin (100 U/mL and 100 µg/mL, respectively), sodium pyruvate, 0.25% trypsin-EDTA, and phosphate-buffered saline (PBS) were obtained from Gibco BRL (Grand Island, NY, USA). All media and supplements were stored and handled according to the manufacturer’s instructions to maintain sterility and performance. Polyvinylidene fluoride (PVDF) membranes and prestained molecular weight protein markers were purchased from Bio-Rad Laboratories (Hercules, CA, USA). The membranes were used for western blotting procedures, and the markers served as molecular weight references during SDS-PAGE analysis. Additional reagents used for protein analysis, including lysis buffers and blocking agents, were either acquired from standard commercial sources or prepared in-house using analytical-grade chemicals.

### 2.2. Cell Culture

ZR-75-1 (CRL-1500, ATCC) and MDA-MB-231 (HTB-26, ATCC) human breast cancer cell lines were obtained from the American Type Culture Collection (ATCC, Manassas, VA, USA). ZR-75-1 cells were cultured in RPMI-1640 medium (Gibco BRL, Grand Island, NY, USA), while MDA-MB-231 cells were maintained in Leibovitz’s L-15 medium (Gibco BRL), according to ATCC recommendations. Both media were supplemented with 10% (*v*/*v*) fetal bovine serum (FBS; HyClone, Logan, UT, USA) and 1% antibiotic solution containing 200 units/mL penicillin and 200 μg/mL streptomycin (HyClone). Cells were cultured in T-75 flasks (Corning, NY, USA) and maintained at 37 °C in a humidified atmosphere. ZR-75-1 cells were incubated in a 5% CO_2_/95% air environment, while MDA-MB-231 cells, due to the CO_2_-independent nature of L-15 medium, were grown in ambient air without supplemental CO_2_. The cells were passaged every 2–3 days using 0.25% trypsin-EDTA (Gibco BRL) once they reached approximately 80–90% confluence. All cultures were routinely examined under an inverted microscope to assess cell morphology and density, and were tested regularly for mycoplasma contamination using a PCR-based detection kit.

### 2.3. Cell Proliferation Assay

Cells were seeded into 96-well culture plates at a density of 5000 cells per well to ensure uniform growth conditions and reproducibility across treatments. MDA-MB-231 cells were exposed to increasing concentrations of Simvastatin—specifically 0, 25, 50, and 75 μM—over a time course of 24, 48, and 72 h. In parallel, ZR-75-1 cells were treated under identical time intervals but with slightly lower Simvastatin concentrations of 0, 12.5, 25, and 50 μM, reflecting their differing sensitivities to the drug. Following the designated treatment periods, 1 mg/mL of MTT dye solution was added to each well and incubated for a minimum of 4 h to allow viable cells to reduce MTT to formazan crystals, a marker of metabolic activity. After incubation, the reaction was terminated by carefully adding dimethyl sulfoxide (DMSO) to solubilize the formazan product. The resulting color intensity, directly proportional to the number of viable cells, was quantified by measuring absorbance at 540 nm using a calibrated multi-well microplate reader. To ensure accurate results, background absorbance from wells containing only medium and reagents (without cells) was measured and subtracted from all readings. Each experimental condition was performed in triplicate within each run to reduce variability, and the experiments were independently repeated at least three times to ensure reliability. The absorbance values were normalized to the untreated control group, which was set to 100%, and results were expressed as a percentage relative to this baseline. Data were presented as the mean ± standard error of the mean (SEM), providing a measure of the precision of the experimental estimates.

### 2.4. Western Blot Assay

A total of 30–50 μg of proteins were separated by SDS-PAGE (10–12% SDS-polyacrylamide gel electrophoresis) and transferred to PVDF membranes (Millipore, Billerica, MA, USA) in a tank blotter (in 25 mM Tris, 0.192 M glycine, pH 8.3, 20% methanol) at 30 volts overnight. The membranes were blocked with 5% non-fat milk (in 10 mM Tris-HCl, pH 8.0, 150 mM NaCl, 0.05% Tween-20) overnight and incubated with anti-β-actin (AC-15 Sigma, St. Louis, MO, USA), anti-CDK4 (SC-601), Cyclin D (SC-753), CDK2 (SC-748), and GSK3β (SC-9166) antibodies (Santa Cruz, USA) for 1.5–2 h. The blots were washed with Tris-HCl (pH 8.0, 150 mM NaCl, 0.05% Tween-20) for 3 × 10 min and incubated with a secondary antibody (anti-rabbit or anti-mouse immunoglobulins) (IRDye Li-COR, Lincoln, NE, USA) at 1:200 dilution for 1 h. The antigen was then visualized and analyzed using the Odyssey infrared imaging system (Odyssey LI-COR, Lincoln, NE, USA).

### 2.5. Evaluation of Apoptosis/Necrosis

Apoptosis and necrosis were quantitatively assessed using the ApopNexin FITC apoptosis detection kit (Chemicon, Temecula, CA, USA), which enables the discrimination between live, early apoptotic, late apoptotic, and necrotic cells based on membrane integrity and phosphatidylserine exposure. MDA-MB-231 and ZR-75-1 breast cancer cells were seeded under standard conditions and treated with various concentrations of Simvastatin for a duration of 6 h to evaluate the early cellular responses to drug exposure. Following treatment, cells were harvested, washed twice with cold phosphate-buffered saline (PBS), and stained with fluorescein isothiocyanate (FITC)-conjugated annexin V and propidium iodide (PI), according to the manufacturer’s instructions. This dual staining method allows for the identification of apoptotic cells (annexin V–positive, PI–negative) and necrotic or late apoptotic cells (annexin V–positive, PI–positive). The stained cells were immediately subjected to flow cytometric analysis using a FacsCalibur cytometer (Becton Dickinson, BD, Franklin Lakes, NJ, USA), which quantitatively measured the fluorescence intensities and enabled classification of cell populations based on viability status. Flow cytometry data were collected and subsequently analyzed using WinMDI version 2.9 (BD, USA), a freely available software tool for multi-parameter cytometric data visualization and analysis. Each condition was assessed in at least three independent experimental replicates to ensure the reproducibility and statistical reliability of the findings. The results provided insights into whether Simvastatin induces apoptosis, necrosis, or both, in a concentration-dependent manner.

### 2.6. Cell Cycle Analysis

For comprehensive cell cycle analysis utilizing propidium iodide (PI) staining, MDA-MB-231 and ZR-75-1 human breast cancer cell lines were subjected to treatment with varying concentrations of Simvastatin, or alternatively, serum-starved for 24 h to induce cell cycle arrest as a comparative control. After the treatment period, cells were trypsinized, collected by centrifugation, and washed twice with cold phosphate-buffered saline (PBS) to remove residual serum and media components. Subsequently, the cell pellets were gently resuspended in 1 mL of cold 70% ethanol, added dropwise while vortexing to prevent clumping, and fixed for a minimum of 8 h at −20 °C to permeabilize cellular membranes and preserve nuclear DNA content. Following fixation, cells were centrifuged to remove ethanol, rehydrated in PBS, and stained with a DNA-specific PI/RNase A solution. This solution, containing propidium iodide and RNase A, enables selective staining of double-stranded DNA while degrading RNA, thus ensuring accurate quantification of DNA content without interference from RNA. The stained cell suspensions were then analyzed by flow cytometry using a FacsCalibur system (Becton Dickinson, BD, USA), and a minimum of 10,000 single-cell events were acquired per sample to ensure statistically meaningful results. Proper gating strategies were applied to exclude cell debris and doublets, allowing for precise discrimination of the G0/G1, S, and G2/M phases based on DNA content. Raw cytometric data were processed and interpreted using WinMDI version 2.9, a freely available software tool from BD Biosciences, which allowed for graphical representation and quantitative analysis of the cell cycle distribution. Each experimental condition was performed in triplicate, and data were presented as mean values with standard error to ensure reproducibility and statistical validity.

### 2.7. Differential Gene Expression Collection

The dataset expression profile of GSE38959, GSE45827, and GSE65194 were obtained from Gene expression omnibus (GEO). Three transcriptomic datasets of breast cancer, along with corresponding clinical data. These transcriptome profiles were pre-processed and analyzed as previously described. Gene expression datasets of TCGA-BRCA was extracted from DriverDBv4 (http://driverdb.bioinfomics.org/ accessed on 23 November 2024) and GEPIA2 (http://gepia2.cancer-pku.cn/#index accessed on 23 November 2024). The gene expression difference of a single gene in the three data sets is defined by log_2_(Fold change) and adjusted *p* value. Among them, log2FC > 1 or <−1 and adjusted *p* value < 0.00001 are defined as common genes. A Venn diagram was performed to illustrate overlapping genes in these datasets.

### 2.8. Statistics

All data were reported as the mean (±SEM) of at least three separate experiments. Statistical analysis was performed using a *t*-test or one-way ANOVA, followed by Turkey’s post hoc test, with significant differences determined at *p* < 0.05.

## 3. Results

### 3.1. Simvastatin Inhibited Cell Survival/Proliferation of TNBC Cells (MDA-MB-231 and ZR-75-1)

In this study, we hypothesized that Simvastatin may influence the survival of TNBC cells by suppressing their proliferative capacity. To evaluate the potential anti-tumor properties of Simvastatin, we designed an in vitro experiment in which TNBC cells were exposed to increasing concentrations of the drug. Specifically, MDA-MB-231 cells were treated with 0, 25, 50, and 75 μM of Simvastatin, while ZR-75-1 cells received 0, 12.5, 25, and 50 μM, with incubation periods ranging from 24 to 72 h. Cell proliferation was quantitatively measured using the MTT assay to assess metabolic activity as an indirect marker of cell viability.

The results, as illustrated in the upper panels of Figure 1A,B, revealed that Simvastatin exerted a dose-dependent inhibitory effect on the survival and proliferation of both MDA-MB-231 and ZR-75-1 cancer cells, particularly notable after 24 h of treatment. In contrast, treatment with equivalent concentrations of Simvastatin did not result in significant viability changes in Hs-68 normal human skin fibroblasts or MRC-5 normal lung fibroblasts, which served as non-cancerous control cell lines, suggesting a degree of selectivity for cancer cells. (Figure 1C,D).

### 3.2. Simvastatin Inhibits Cell Proliferation

To further explore the role of Simvastatin in modulating apoptosis in triple-negative breast cancer (TNBC) cells, we performed flow cytometric analysis using Annexin V-FITC and propidium iodide (PI) dual staining, a widely accepted method for detecting early and late apoptotic events. MDA-MB-231 and ZR-75-1 cells were exposed to Simvastatin for 24 h, after which apoptotic populations were quantified and compared to untreated and DMSO-treated control groups (Figure 2A). The results in Figure 2B are expressed as the percentage of total apoptotic cells, encompassing both early and late apoptotic populations. The resulting dot plots of Annexin V-FITC versus PI fluorescence revealed that Simvastatin did not significantly increase the proportion of apoptotic MDA-MB-231 cells, as the levels remained comparable to those observed in both control conditions.

Similarly, no statistically significant increase was detected in necrotic or apoptotic ZR-75-1 cells upon Simvastatin treatment, a key executor of apoptosis. However, a modest elevation in the percentage of apoptotic ZR-75-1 cells was observed when compared to the untreated and DMSO groups, indicating a partial or cell line–specific response to Simvastatin.

Collectively, the data shown in Figure 1 and Figure 2A,B suggest that while Simvastatin exerts a measurable impact on the survival of MDA-MB-231 and ZR-75-1 cells, this effect is likely independent of the canonical apoptosis pathways. Therefore, we propose that the antiproliferative action of Simvastatin in TNBC occurs primarily through alternative, apoptosis-independent mechanisms.

### 3.3. Simvastatin Induces Apoptosis in Non-TNBC (ZR-75-1) Cells but Not in TNBC (MDA-MB-231) Cells

To investigate the influence of Simvastatin on cell cycle regulation in triple-negative breast cancer (TNBC) cells, we performed cell cycle analysis using flow cytometry following treatment with Simvastatin. Cells were incubated with varying concentrations of Simvastatin for 24 h prior to sample processing and staining with propidium iodide (PI) to assess DNA content. Flow cytometric profiles were then generated to evaluate the distribution of cells across different cell cycle phases.

As shown in Figure 3A,B, Simvastatin induced an increase in G1 phase and a decreased in S phase cellsSimvastatin treatment resulted in a noticeable shift in cell cycle distribution, characterized by a marked increase in the proportion of cells in the G1 phase. Concurrently, a reduction was observed in the number of cells residing in both the S and G2/M phases. Figure 3A shows MDA-MB-231 and ZR-75-1 cells after treatment with Simvastatin for 24 h. Figure 3B demonstrates that Simvastatin treatment led to an increase in the proportion of cells in the G1 phase and a decrease in the S phase. Concurrently, a reduction was observed in the number of cells residing in both the S and G2/M phases. This redistribution pattern suggests that Simvastatin induces a G1 phase arrest in TNBC cells, thereby impeding progression through the cell cycle. Furthermore, this effect was found to be dose-dependent, with higher concentrations of Simvastatin producing a more pronounced accumulation of cells in the G1 phase. Statistical analysis confirmed the significance of this observation, as the percentage of G1-arrested cells was significantly elevated in treated groups compared to the untreated control (* *p* < 0.05, ** *p* < 0.01, ****p* < 0.001 vs. Simvastatin 0 μM). These findings strongly support the hypothesis that Simvastatin exerts its antiproliferative effects, at least in part, by halting cell cycle progression at the G1 checkpoint.

### 3.4. Protein Expression of Cell Cycle Progression Following Exposure to Simvastatin

The immunoblotting results for cellular proteins from MDA-MB-231 and ZR-75-1 cells treated with Simvastatin provide important insights into the molecular effects of this drug. Specifically, in TNBC cells (MDA-MB-231), we observed that Cyclin D1 expression was downregulated at a lower dose (25 μM), whereas CDK4 was significantly suppressed at a higher dose (75 μM). In contrast, in non-TNBC cells (ZR-75-1), CDK4 downregulation was primarily observed at the higher dose (75 μM) (Figure 4A,B). Accordingly, we have revised the corresponding paragraph in the Results section to reflect these findings more accurately. The updated interpretation emphasizes that Simvastatin may differentially modulate key G1/S checkpoint proteins in a dose-dependent and cell type-specific manner. These results further support the hypothesis that CDK4 is a more consistently responsive target, particularly at higher concentrations, while Cyclin D1 modulation appears to be more sensitive to lower doses in TNBC cells. The observed increase in the proportion of cells in the G1 phase is likely attributable to the non-apoptotic action of Simvastatin, particularly its ability to downregulate CDK4 expression. This mechanism appears to operate independently of apoptotic pathways and instead highlights a specific regulatory effect on cell cycle progression.

### 3.5. High Expression of CDK4 Was Strongly Associated with the Prognosis of BRCA

To further investigate the pivotal role of CDK4 expression in breast cancer (BRCA), we conducted an in-depth analysis by screening three gene expression datasets—GSE38959, GSE45827, and GSE65194—obtained from the Gene Expression Omnibus (GEO) database. As shown in Figure 5A, a Venn diagram was generated to identify overlapping differentially expressed genes (DEGs) across these datasets. Notably, a total of 949 genes were found to be commonly dysregulated among BRCA and normal breast tissue samples, with CDK4 being one of the key overlapping genes identified. Following this, we performed a more detailed comparative analysis of the gene expression profiles in each dataset. Specifically, the GSE38959 dataset comprised 30 TNBC samples and 13 normal breast specimens; the GSE45827 dataset included 34 TNBC samples and 11 normal controls; and the GSE65194 dataset contained 55 TNBC samples alongside 11 normal breast tissue samples. The analysis of these datasets consistently revealed that CDK4 expression was significantly upregulated in TNBC tissues compared to normal breast tissues (Figure 5B), highlighting a strong association between elevated CDK4 expression and TNBC.

To further assess the potential clinical relevance of CDK4 in BRCA, we conducted Kaplan–Meier survival analyses using tools from DriverDBv4 and GEPIA2 platforms. As illustrated in Figure 5C, the results indicated that BRCA patients exhibiting higher levels of CDK4 expression had significantly poorer overall survival (OS) outcomes than those with lower expression levels. This finding suggests that CDK4 overexpression may serve as a prognostic indicator linked to unfavorable clinical outcomes in breast cancer patients. Moreover, validation using The Cancer Genome Atlas (TCGA) datasets further supported these observations. Specifically, analyses via DriverDBv4 and GEPIA2 confirmed that CDK4 expression was markedly elevated in BRCA tissues relative to normal breast tissues (Figure 5D), reinforcing the hypothesis of CDK4’s oncogenic role.

CDK4 was not only significantly overexpressed in TNBC and BRCA samples but also closely associated with adverse clinical prognosis. These results collectively identify CDK4 as a potential oncogenic driver and promising biomarker candidate in TNBC, warranting further investigation into its functional role and therapeutic relevance.

## 4. Discussion

Simvastatin effectively inhibits the survival and proliferation of TNBC, primarily through the induction of apoptosis. Additionally, Simvastatin induces G1 phase cell cycle arrest, suggesting its regulatory effect on cell cycle progression. Further analysis revealed that Simvastatin specifically reduces the expression of CDK4, a key cell cycle regulator. Notably, high CDK4 expression is strongly associated with poor prognosis in breast carcinoma, indicating that CDK4 may serve as a critical target in the anticancer mechanism of Simvastatin.

Statins have long served as the foundation of pharmacologic intervention in cardiovascular disease, primarily by inhibiting HMG-CoA reductase to lower serum cholesterol levels and reduce atherosclerotic risk [15]. However, accumulating evidence indicates that the benefits of statins extend well beyond cardiovascular protection. Simvastatin has emerged as a leading candidate among statins with potential pleiotropic effects, including anti-inflammatory, immunomodulatory, and notably, anticancer properties [16]. These anticancer effects appear to be mediated through mechanisms independent of cholesterol regulations such as the inhibition of the mevalonate pathway’s downstream products involved in cell growth and survival [9]. Among various malignancies, TNBC presents an urgent therapeutic challenge due to its aggressive nature and lack of targeted therapies. The observation that Simvastatin can suppress CDK4 expression and arrest TNBC cells in the G1 phase of the cell cycle further underscores its potential utility in oncology. Moreover, the repurposing of Simvastatin is particularly relevant in the clinical setting where a substantial subset of breast cancer patients—especially postmenopausal women—also present with hyperlipidemia. For these patients, Simvastatin offers a dual therapeutic advantage by concurrently managing lipid levels and exerting direct anticancer effects. This duality not only enhances the clinical value of Simvastatin but also supports a cost-effective, well-tolerated strategy for addressing complex comorbid conditions in cancer care.

Inducing apoptosis is a fundamental strategy in cancer therapy, as it promotes the selective elimination of malignant cells while sparing normal tissue. In this study, Simvastatin effectively inhibited the survival and proliferation of BCCs, particularly in MDA-MB-231 and ZR-75-1 lines, in a dose-dependent manner over a 24-h treatment period (Figure 1). Importantly, Simvastatin had a minimal impact on normal fibroblast cells, such as Hs-68 and MRC-5, suggesting a degree of cancer cell specificity. Morphological changes observed in treated cancer cells—such as detachment and rounding—are commonly associated with apoptotic processes. Although initial analyses suggested that the reduction in proliferation might occur independently of apoptosis (Figure 2), the overall evidence, including cell viability loss and structural alterations, supports the conclusion that Simvastatin exerts at least part of its anticancer effect by promoting apoptotic cell death. These findings are consistent with previous studies reporting that statins, including Simvastatin, can induce apoptosis in various cancer types through pathways involving caspase activation, mitochondrial dysfunction, and cell cycle arrest [10,17,18]. These results highlight Simvastatin’s potential as a selective therapeutic agent for TNBC, capable of not only arresting growth but also activating apoptotic pathways in cancer cells.

Simvastatin, a well-known cholesterol-lowering drug, has demonstrated promising anti-cancer properties, particularly against TNBC, a subtype lacking targeted therapies. In our study, simvastatin significantly suppressed TNBC cell proliferation via a non-apoptotic mechanism involving G1 cell cycle arrest and CDK4 downregulation. This finding adds a new dimension to simvastatin’s anticancer action and complements existing research on its molecular targets. Wang et al. (2016) [10] showed that simvastatin inhibits the PI3K/Akt and MAPK/ERK pathways by dephosphorylating c-Raf, MEK1/2, and ERK1/2, effects that are reversed by metabolites of the mevalonate pathway. Park et al. [19] further confirmed simvastatin’s ability to suppress Akt phosphorylation, particularly in PTEN-deficient TNBC cells. Kou et al. (2018) [20] identified simvastatin as a novel heat shock protein 90 (Hsp90) inhibitor, disrupting the Hsp90/Cdc37 complex and leading to cancer cell death. Additionally, Wolfe et al. (2015) [12] reported that simvastatin suppresses metastasis via FOXO3a regulation, and Bai et al. (2019) [21] linked simvastatin-induced oxidative stress to the upregulation of miR-140-5p, contributing to cancer cell death. Taken together, these studies underscore simvastatin’s multifaceted anti-tumoral activities, including inhibition of proliferation, survival signaling, chaperone activity, and metastasis. Our findings further emphasize its ability to induce cell cycle arrest via CDK4 suppression. Given simvastatin’s established clinical safety and widespread use, it holds potential as a cost-effective adjunct in TNBC treatment, particularly for patients with comorbid dyslipidemia.

Emerging clinical evidence supports a potential role for statins in improving outcomes among patients with TNBC. In a large retrospective cohort study, Bai et al. (2019) [21] analyzed data from over 2000 TNBC patients and found that statin use was significantly associated with improved overall survival and breast cancer–specific survival, particularly among patients with high-grade tumors. These findings suggest that statins may exert beneficial effects beyond lipid-lowering, potentially through modulation of tumor biology, inflammation, and cellular signaling pathways. The study also highlighted that lipophilic statins, such as simvastatin, were more strongly associated with improved outcomes, supporting their potential therapeutic role in TNBC. Our current findings align with this clinical evidence and offer mechanistic insight into how simvastatin may contribute to TNBC growth suppression via CDK4 downregulation and cell cycle arrest. Taken together, these data provide a compelling rationale for further clinical evaluation of statins, particularly simvastatin, as adjunctive agents in the management of TNBC.

Inducing cell cycle arrest, especially at the G1 phase, is a crucial strategy in cancer therapy, as it can halt uncontrolled cell proliferation and provide opportunities for therapeutic intervention [22]. In this study, Simvastatin treatment induced a significant accumulation of TNBC in the G1 phase (Figure 3), with a corresponding decrease in the S and G2 phases, demonstrating a dose-dependent induction of G1 phase arrest in TNBC cells. The results suggest that Simvastatin may cause G1 phase arrest through mechanisms independent of apoptosis [23,24]. Specifically, Simvastatin reduced the expression of CDK4, a key regulator of the G1-S transition, which is involved in the progression from the G1 phase to the S phase. This finding aligns with previous studies indicating that statins, including Simvastatin, can induce G1 phase arrest by inhibiting CDK4 and other cell cycle-related proteins. This non-apoptotic mechanism is particularly relevant for therapeutic strategies targeting cell cycle progression in cancer cells, where CDK4 overexpression has been linked to poor prognosis and increased cell proliferation in breast cancer [25,26,27]. Together, these results underscore the potential of Simvastatin as a therapeutic agent for TNBC by selectively targeting G1 phase arrest and CDK4 expression to inhibit tumor cell growth.

Cyclin-dependent kinase 4 (CDK4) plays a crucial role in regulating the G1 to S phase transition of the cell cycle by forming an active complex with cyclin D, which phosphorylates the retinoblastoma protein (Rb) [28], thereby promoting E2F-mediated transcription and DNA synthesis [27]. Dysregulation of this pathway, particularly through CDK4 overexpression, contributes to unchecked cellular proliferation and is frequently associated with poor prognosis in various cancers, including breast carcinoma [29]. Our data indicate that CDK4 downregulation is a consistent and prominent effect of Simvastatin treatment, particularly at higher concentrations (75 μM), in both TNBC (MDA-MB-231) and non-TNBC (ZR-75-1) cell lines. This effect was not paralleled by equally significant changes in other G1/S regulatory proteins, such as Cyclin E or CDK2.Furthermore, while Cyclin D1 showed a reduction in expression at lower concentrations (25 μM) in MDA-MB-231 cells, this effect was less consistent and more dose-sensitive compared to CDK4. Therefore, we clarified in the revised manuscript that Simvastatin appears to exert a relatively more selective and robust inhibitory effect on CDK4, rather than broadly suppressing all components of the G1/S checkpoint (Figure 4). This effect coincided with a reduction in CDK4 protein expression, indicating that Simvastatin-induced G1 arrest may occur via non-apoptotic mechanisms involving suppression of CDK4. Notably, high CDK4 expression has been strongly correlated with worse outcomes in TNBC, supporting its identification as a potential oncogene and therapeutic target. The ability of Simvastatin to downregulate CDK4 highlights a novel mechanism by which it may exert anticancer activity in TNBC cells. This observation is consistent with previous studies reporting that CDK4 inhibition suppresses tumor growth and enhances therapeutic sensitivity [23,30]. This effect coincided with a reduction in CDK4 protein expression, suggesting that Simvastatin-induced G1 arrest may involve suppression of CDK4. However, this association remains correlative, and further mechanistic studies—such as CDK4 overexpression or knockdown experiments—are warranted to determine whether CDK4 downregulation plays a causative role in Simvastatin-mediated cell cycle arrest. Collectively, our findings support the potential of Simvastatin as a candidate for targeting CDK4 to induce G1 arrest and ultimately promote apoptosis in breast cancer cells. Our results indicate that simvastatin induces G1 cell cycle arrest in TNBCs through the downregulation of CDK4, suggesting that Simvastatin’s anti-cancer effects are associated with its ability to suppress the cell cycle. This finding underscores the anti-carcinogenic and protective role of Simvastatin in ZR-75-1 cells. However, as illustrated in Figure 1 and Figure 3, Simvastatin appears to affect the survival of both MDA-MB-231 and ZR-75-1 cells through mechanisms other than apoptosis. Therefore, the inhibition of cell proliferation by Simvastatin in these cell lines is likely mediated by pathways distinct from those involving apoptosis.

Recent studies have highlighted the critical role of CDK4/6 in the regulation of cell cycle progression, particularly in breast cancer. While CDK4/6 inhibitors such as palbociclib have demonstrated significant clinical efficacy in hormone receptor-positive (HR+) breast cancer, their role in TNBC remains under active investigation. Some reports suggest that a subset of TNBC exhibits elevated CDK4/6 activity and may be sensitive to CDK4/6 inhibition [31]. These findings underscore the potential of CDK4/6 as therapeutic targets and prognostic biomarkers in TNBC, warranting further mechanistic and clinical validation.

Simvastatin induced G1 phase cell cycle arrest in MDA-MB-231 and ZR-75-1 breast cancer cells, which was associated with the downregulation of CDK4. This suggests a mechanism by which Simvastatin suppresses cell cycle progression through interference with the CDK4-cyclin D axis. The PI3K/GSK3β signaling axis further intersects with this regulation, as GSK3β negatively controls cyclin D1 by promoting its phosphorylation and degradation, indirectly reducing CDK4/6 activation. Thus, Simvastatin’s modulation of GSK3β and cyclin D1 stability may contribute to the inhibition of CDK4-driven cell cycle progression. Notably, CDK4 overexpression is frequently implicated in breast carcinoma progression, and pharmacological inhibition of CDK4/6 has demonstrated clinical efficacy in suppressing tumor proliferation [23,30].

Interestingly, our findings suggest that G1 cell cycle arrest was observed primarily at three different Simvastatin doses in MDA-MB-231 and ZR-75-1 cells, while apoptosis and caspase activation were more prominent at lower doses in NSCLC6 and other cancer cell lines [32,33]. This analysis led us to conclude that both the inhibition of proliferation and the induction of apoptosis and cell cycle arrest are dependent on the specific type of cancer cell. Further investigation indicated a more complex mechanism involving cell cycle deregulation and apoptosis, which appears to vary with the degree of Simvastatin-induced toxicity among different cancer cell lines.

This study has several limitations that should be considered when interpreting the results. Specifically, we have highlighted those in vitro results using a single TNBC cell line (MDA-MB-231) may not fully represent the heterogeneity of TNBC in clinical settings, and those in vivo studies are necessary to evaluate Simvastatin’s efficacy, bioavailability, and safety in a more physiologically relevant environment. Furthermore, although Simvastatin was found to induce G1 phase arrest, it did not significantly promote apoptosis or necrosis within the tested timeframe, suggesting a primarily cytostatic effect that may limit its standalone therapeutic impact. To further validate these findings, we assessed mitochondrial membrane potential—an early indicator of apoptosis and upstream trigger of the caspase cascade—using JC-1 dye staining. The flow cytometry results demonstrated that Simvastatin treatment did not result in a significant decrease in mitochondrial membrane potential in either cell line, supporting the hypothesis that classical apoptotic signaling is not the predominant mechanism of Simvastatin-induced growth inhibition in TNBCs.

We fully agree that further validation is critical. As part of our ongoing and future studies, we plan to extend this investigation to cisplatin- and doxorubicin-resistant TNBC cell lines, as well as to establish xenograft models in immunocompromised mice to assess in vivo efficacy and pharmacodynamics of simvastatin treatment. These models will help clarify whether the observed CDK4-mediated G1 arrest remains effective under chemoresistant conditions and within the tumor microenvironment. We are currently planning follow-up in vivo experiments using TNBC xenograft models to further investigate the therapeutic potential, pharmacodynamics, and possible off-target effects of Simvastatin in vivo. The relatively high IC50 value of Simvastatin observed in vitro represents a potential limitation in terms of its clinical applicability. While this may suggest a need for high concentrations to achieve therapeutic efficacy, it is important to consider the broader pharmacological context. Simvastatin’s mechanism of action as an HMG-CoA reductase inhibitor—particularly its impact on the downstream metabolites of the mevalonate pathway (such as farnesyl pyrophosphate and geranylgeranyl pyrophosphate), its regulation of CDK4 expression, and its influence on mitochondrial function—remains of significant biological importance. These interconnected mechanisms not only contribute to the observed antiproliferative and pro-apoptotic effects, but also provide a rationale for combination strategies or drug delivery approaches to enhance its potency. Taken together, these aspects help to deepen the understanding of Simvastatin’s anticancer potential and strengthen the mechanistic interpretation of our findings. Finally, the study’s observation period was limited to 72 h, which does not account for the long-term effects of Simvastatin on cell viability, resistance development, or delayed cytotoxicity, all of which warrant future investigation.

Although the concentrations of Simvastatin used in this study (25–75 μM) are higher than the plasma levels typically achieved in patients receiving standard oral doses (0.01–0.1 μM due to extensive first-pass metabolism and high plasma protein binding), such concentrations are commonly applied in in vitro cancer research. This approach compensates for the lack of metabolic processes and bioavailability constraints in cell culture systems. Importantly, similar in vitro concentrations have been widely used in previous studies investigating the anticancer effects of statins across various cancer types, including breast cancer [34]. These higher concentrations are also helpful in uncovering pleiotropic, non-lipid-lowering effects of statins that may not manifest at lower doses. While in vivo validation and pharmacokinetic modeling will be required to confirm translational potential, the present findings provide a mechanistic foundation for future studies at more clinically relevant exposures.

## 5. Conclusions

This study provides compelling evidence that Simvastatin exerts anticancer effects against TNBC cells by inducing both apoptosis and G1 phase cell cycle arrest. Mechanistically, Simvastatin downregulates CDK4 expression, a pivotal regulator of G1/S transition whose overexpression is associated with poor prognosis in breast cancer. These findings underscore the potential of Simvastatin as a therapeutic agent targeting CDK4-driven cell cycle dysregulation in TNBC. Future studies, including in vivo validations and clinical investigations, are warranted to further substantiate these findings.

## Figures and Tables

**Figure 1 cimb-47-00622-f001:**
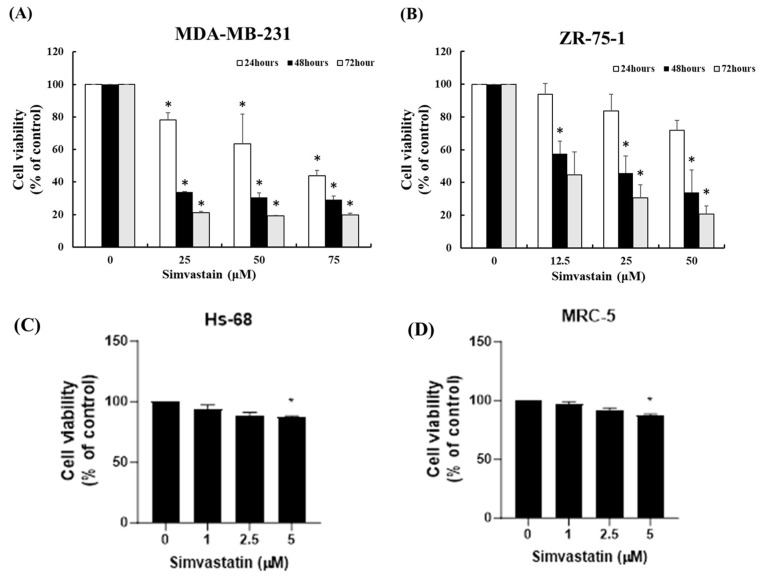
Simvastatin inhibited cell survival/proliferation of TNBC cells. (**A**) Simvastatin mediates the survival of MDA-MB-231 and (**B**) ZR-75-1 cells (*n* = 3 per group) and thus inhibits their proliferation. In vitro study was initiated by treating each cell line with increasing doses of Simvastatin for 24 h. The survival of these Simvastatin-treated MDA-MB-231 and ZR-75-1 cells was then measured by MTT method. (**C**,**D**) showing MTT assay results for normal human fibroblast cell lines HS68 (skin) and MRC5 (lung) treated with varying concentrations of Simvastatin. Results were expressed as a percentage of control, which was considered as 100%. All data were reported as the means (±SEM) of at least three separate experiments. Statistical analysis used the *t*-test, with the significant differences determined at the level of * *p* < 0.05 versus control (0 uM) or time 0 group.

**Figure 2 cimb-47-00622-f002:**
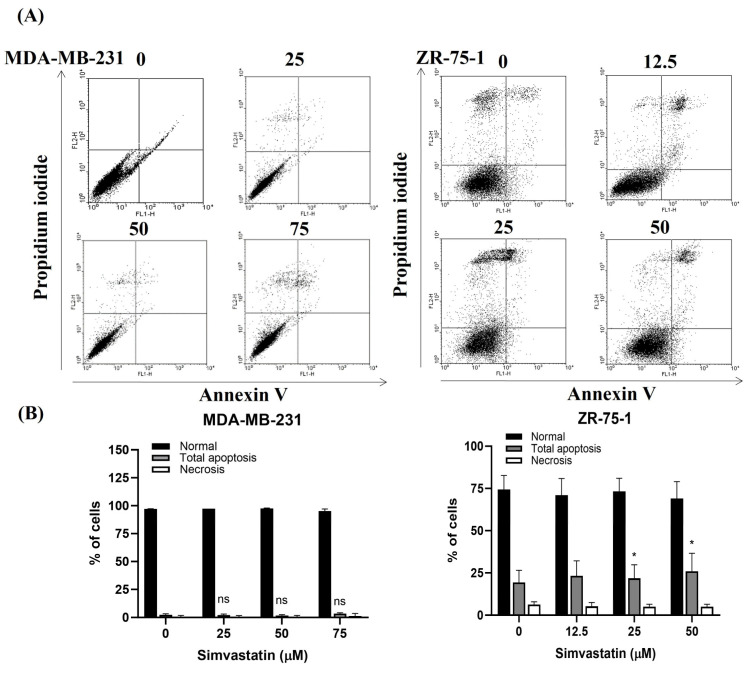
Influence of Simvastatin on TNBCs. (**A**) Total apoptosis and necrosis in MDA-MB-231 and ZR-75-1 cells after 4 h of incubation with Simvastatin. Simvastatin did not significantly increase the apoptosis rate of MDA-MB-231 cells, but on the contrary, it increased the apoptosis rate of ZR-75-1 cells. (**B**) Results were expressed as a percentage of control, which was considered as 100%. All data were reported as the means (±SEM) of at least three separate experiments. Statistical analysis used the *t*-test, with the significant differences determined at the level of * *p* < 0.05 versus control (0 uM) or time 0 group.

**Figure 3 cimb-47-00622-f003:**
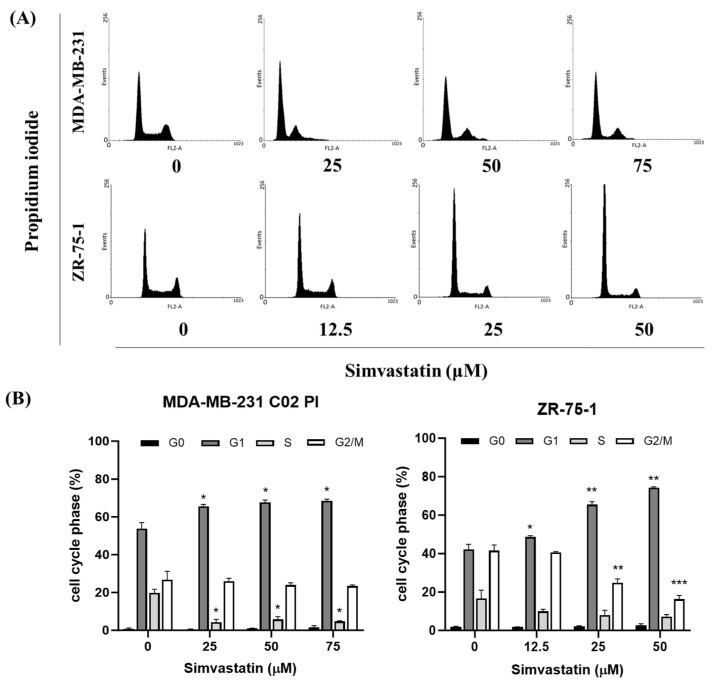
Influence of Simvastatin on cell cycle progression/distribution in TNBCs: Cell cycle analysis of (**A**) MDA-MB-231 and ZR-75-1 cells after being cultured with Simvastatin for 24 h. (**B**) Results were expressed as a percentage of control, which was considered as 100%. Simvastatin treatment resulted in an increase in the proportion of cells in the G1 phase and a decrease in the proportion of cells in the S phase. In addition, a decrease in the number of cells in the S and G2/M phases was also observed. All data were reported as the means (±SEM) of at least three separate experiments. Statistical analysis used the *t*-test, with the significant differences determined at the level of * *p* < 0.05, ** *p* < 0.01, *** *p* < 0.001 versus control (uM) or time 0 group.

**Figure 4 cimb-47-00622-f004:**
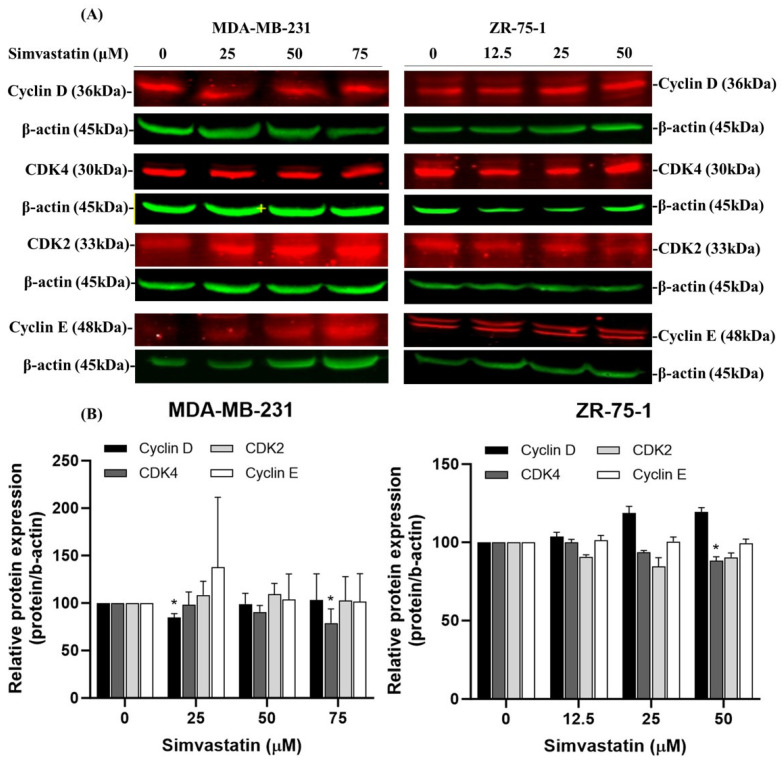
Simvastatin represses CDK4 protein expression in TNBCs. (**A**) MDA-MB-231 and ZR-75-1 cells were treated with Simvastatin (0, 25, 50 and 75 μM) and (0, 12.5, 25 and 50 μM) for 24 h. (**B**) The protein expressions were subsequently detected CDK4, CDK2, Cyclin D1, and Cyclin E1 proteins (all red colors) by Western blot analysis. Quantification of band intensities was shown in lower portion. Results were expressed as a percentage of control, which was considered as 100%. All data were reported as the means (±SEM) of at least three separate experiments. Statistical analysis used the *t*-test, with the significant differences determined at the level of * *p* < 0.05 versus control (0 uM) or time 0 group. β-actin was used as a loading control (green color).

**Figure 5 cimb-47-00622-f005:**
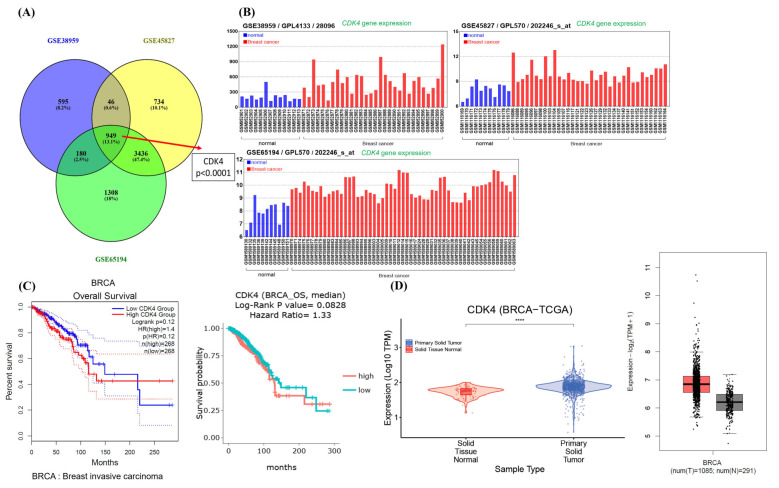
CDK4 is highly expressed in BRCA tissues and is closely related to poor prognosis. (**A**) Venn diagram for both three datasets of comparisons, indicating CDK4 was one of the most significantly involved in BRCA tissues and normal breast specimens. (**B**) Gene expression analysis with GSE38959, GSE45827, and GSE65194 datasets in CDK4 expression in BRCA tissues and normal tissues. (**C**) Kaplan-Meier analysis for CDK4 expression using BRCA patient samples. (**D**) CDK4 gene expression analysis using TCGA BRCA database. High expression of CDK4 in BRCA tissues, compared with normal tissue. **** *p* < 0.0001 versus solid tissue normal with primary solid tumor.

## Data Availability

The data used to support this study are available upon request from the corresponding authors.

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
