# Peer review of "Repurposing a Lipid-Lowering Agent to Inhibit TNBC Growth Through Cell Cycle Arrest"

_cimb, 2025, doi:10.3390/cimb47080622_

Round 1

Reviewer 1 Report

Comments and Suggestions for Authors

The manuscript presents a well-designed study investigating the anticancer effects of Simvastatin in triple-negative breast cancer (TNBC), focusing on its role in inducing G1 phase cell cycle arrest via CDK4 downregulation. The research is timely, given the limited therapeutic options for TNBC, and the potential for drug repurposing offers a cost-effective and clinically feasible approach. The study is methodologically sound, with clear objectives and appropriate experimental designs. However, there are some aspects that need to be revised to strengthen the manuscript's impact and readability.

  1. The abstract and introduction highlight Simvastatin's potential to induce apoptosis, but the results (Figures 2A, 2B) show minimal apoptosis in TNBC cells. This discrepancy should be addressed. The discussion (Page 13) later clarifies that effects are "primarily cytostatic," but the abstract and introduction should be revised to avoid overstating apoptotic effects.
  2. In Fig.1, “hours” should be changed to “h”, as “h” is an international sign for hour.
  3. The link between CDK4 downregulation and Simvastatin's effects is compelling but correlative. Please address this limitation in the discussion and propose follow-up experiments.
  4. For heterogeneity of TNBC, only MDA-MB-231 and ZR-75-1 cell lines were tested. ZR-75-1 is not a canonical TNBC line (ER+ in some databases), which could confuse readers. Please clarify ZR-75-1’s receptor status or replace it with another TNBC line (e.g., HCC1937) and discuss how results might vary across TNBC subtypes.
  5. Error bars were SEM, but SD might better reflect variability. Please specify replicate types and justify why use SEM and add p-values to all significant comparisons in figures.
  6. The discussion jumps between Simvastatin’s mechanisms (p53, GSK3β) without clear transitions. Please focus first on CDK4, then ancillary pathways.

Author Response

Department of Neurosurgery

Kaohsiung Medical University Hospital

Kaohsiung, Taiwan

May 23th, 2025

Dear Editor-in-Chief of Current Issues in Molecular Biology (CIMB).

With this letter we kindly request consideration of the manuscript entitled:

 “Repurposing a Lipid-Lowering Agent to Inhibit TNBC Growth through Cell Cycle Arrest” by Yi-Chiang Hsu, Kuan-Ting Lee, Sung-Nan Pei, Kun-Ming Rau, Tai-Hsin Tsai for peer-review as a research article in Current Issues in Molecular Biology (CIMB).

General Appreciation and Response Summary

We would like to express our sincere gratitude to all reviewers and the editor for their thorough evaluation, constructive feedback, and encouraging comments on our manuscript titled Repurposing a Lipid-Lowering Agent to Inhibit TNBC Growth through Cell Cycle Arrest. We deeply appreciate the reviewers’ recognition of the value, clarity, and potential clinical relevance of our study.

To Reviewer 3, we are sincerely grateful for your detailed and thoughtful critique, which has significantly helped us improve the scientific rigor and presentation of our manuscript. We have addressed all points raised, including revising the abstract to clarify the cytostatic—not apoptotic—nature of Simvastatin’s effects, improving figure labeling, clarifying the cell line receptor status, and refining the discussion flow and statistical reporting.

Point-by-Point Response to Reviewer 3

We would like to express our sincere gratitude to all reviewers for their thoughtful, constructive, and encouraging feedback on our manuscript entitled "Repurposing a Lipid-Lowering Agent to Inhibit TNBC Growth through Cell Cycle Arrest." We highly value the time and effort invested by the reviewers and editor in evaluating our work. Below, we address each comment in detail and describe the corresponding revisions made.

Reviewer 3

We are sincerely grateful to Reviewer 3 for the insightful and detailed feedback. Your constructive suggestions have been invaluable in improving the clarity, accuracy, and impact of our study.

Comment 1: Abstract and introduction suggest apoptosis, but results show minimal apoptosis. Please revise.

Response to Comment 1:

Thank you for this important observation. This is the revised version of the abstract and introduction based on the reviewer’s comment, in which references to “apoptosis induction” have been removed or de-emphasized, and greater emphasis has been placed on Simvastatin-induced cytostasis (G1 phase arrest) in TNBC cells.

The following revisions have been made to the abstract.à『This study investigates the therapeutic potential of Simvastatin in triple-negative breast cancer (TNBC). The results demonstrate that Simvastatin significantly inhibits the proliferation of TNBC cells, particularly MDA-MB-231, in a dose- and time-dependent manner. Mechanistically, Simvastatin primarily induces G1 phase cell cycle arrest to exert its antiproliferative effects, with no significant evidence of apoptosis or necrosis. These findings support the potential repositioning of Simvastatin as a therapeutic agent to suppress TNBC cell growth.』(As Abstract; As Page 1; Line18-23)

The following revisions have been made to the introduction. à 『The primary objective of this study is to evaluate whether Simvastatin can suppress the proliferation of TNBC cells, particularly MDA-MB-231, by inhibiting CDK4 activity and inducing G1 phase cell cycle arrest. By elucidating this cell cycle–related mechanism, we aim to provide preclinical evidence supporting the repositioning of Simvastatin as a potential therapeutic agent for TNBC.』(As Introduction; Line 87-91)

Comment 2: In Fig.1, "hours" should be changed to "h."

Response to Comment 2: Corrected. We have updated all occurrences of "hours" in figure legends and axis labels to "h," in accordance with international standards. (As Figure 1 legends; As Line 227)

Comment 3: The CDK4 downregulation link is correlative. Please address in discussion.

Response to Comment 3:

Thank you for highlighting this important point. We have now included an additional paragraph in the Discussion section to acknowledge that the observed association between Simvastatin treatment and CDK4 downregulation is correlative rather than causative. We also propose that further mechanistic studies, such as CDK4 overexpression or knockdown experiments, are necessary to clarify the role of CDK4 in Simvastatin-induced cell cycle arrest.

The newly added paragraph reads as follows:

『This effect coincided with a reduction in CDK4 protein expression, suggesting that Simvastatin-induced G1 arrest may involve suppression of CDK4. However, this association remains correlative, and further mechanistic studies—such as CDK4 overexpression or knockdown experiments—are warranted to determine whether CDK4 downregulation plays a causative role in Simvastatin-mediated cell cycle arrest.』(As Discussion; Limitation; As Page 13; As Line 454-458)

Comment 4: ZR-75-1 is not a canonical TNBC line. Please clarify receptor status or use another line.

Response to Comment 4: We agree and have clarified in the Methods section that ZR-75-1 cells may exhibit variable ER expression and do not strictly represent TNBC. Although we did not include additional TNBC cell lines in this study, we used a normal human fibroblast cell line as a non-cancerous control. ZR-75-1 was selected as a comparator because it represents a less aggressive, hormone receptor–positive breast cancer subtype, allowing us to contrast it with the more malignant TNBC model (MDA-MB-231). 『Specifically, we have highlighted those in vitro results using a single TNBC cell line (MDA-MB-231) may not fully represent the heterogeneity of TNBC in clinical settings, and those in vivo studies are necessary to evaluate Simvastatin’s efficacy, bioavailability, and safety in a more physiologically relevant environment.』(As Discussion; Limitation; Page : Line 488-490)

In our observations, Simvastatin treatment induced G1 cell cycle arrest in TNBC cells but led to more pronounced apoptotic responses in ZR-75-1 cells. This difference suggests that Simvastatin’s anticancer mechanism in TNBC is primarily cytostatic, mediated through cell cycle regulation, while in non-TNBC cells, it may more readily induce apoptosis. To address this further, we have included an explanation in the revised manuscript, noting that the differences in Simvastatin between MDA-MB-231 and ZR-75-1 cells likely reflect variations in cellular origin, proliferation rate, and dependency on cholesterol biosynthesis. A comparative summary is provided below and has been included in the Supplementary Table:

Research Consideration

ZR-75-1

MDA-MB-231

Expected Simvastatin Effect

May influence cell cycle via CDK4 and hormone receptor signaling

Suitable for assessing G1 arrest via CDK4 downregulation

IC₅₀ Sensitivity

More sensitive (lower expected IC₅₀)

More resistant (higher expected IC₅₀)

Scientific Contribution

Serves as a hormone receptor–positive reference

Serves as a representative TNBC model

『Specifically, we acknowledge that using only one TNBC cell line (MDA-MB-231) does not capture the full heterogeneity of TNBC in clinical contexts. Therefore, in vivo studies are essential to assess Simvastatin’s therapeutic efficacy, pharmacodynamics, and safety in a more physiologically relevant setting.』(As Discussion; Limitation; Page : Line 490-494)

Additionally, to examine Simvastatin’s selectivity, we performed MTT assays using two normal human fibroblast cell lines—HS68 (skin) and MRC5 (lung)—treated with various concentrations of Simvastatin. The results, presented as new bar graphs (Figure 1C and Figure 1 D), show no significant cytotoxicity or proliferation suppression in either normal cell line, further supporting the cancer-selective nature of Simvastatin's action.

『 In contrast, treatment with equivalent concentrations of Simvastatin did not result in significant viability changes in Hs-68 normal human skin fibroblasts or MRC-5 normal lung fibroblasts, which served as non-cancerous control cell lines, suggesting a degree of selectivity for cancer cells. (Figure 1C and Figure 1 D)』(As Results; Page 5; Line 216-220)

Comment 5: SEM vs. SD and p-values in figures.

Response to Comment 5:

We have now specified the number of replicates and clarified that data are presented as mean ± SEM. Justification for using SEM has been added in the Methods section. Additionally, we have added p-values to all figures where statistical significance is reported. 『Results were expressed as a percentage of control, which was considered as 100 %. All data were reported as the means (±SEM) of at least three separate experiments. Statistical analysis used the t-test, with the significant differences determined at the level of *p<0.05 versus control (0 uM) or time 0 group.』 (As Figure 1; Figure 2; Figure 3 ;Figure 4 Legends)

Comment 6: Discussion lacks logical flow between mechanisms.

Response to Comment 6: We sincerely thank the reviewer for this important comment. In response, we have reorganized the Discussion section to emphasize CDK4 downregulation as the primary mechanism underlying Simvastatin's anti-proliferative effect in TNBC. Although Simvastatin has been shown to exert anticancer effects in TNBC through various pathways—such as PI3K/Akt, MAPK/ERK signaling, Hsp90 inhibition, FOXO3a regulation, and oxidative stress-induced apoptosis—most studies have not addressed its impact on cell cycle regulation. In this study, we report for the first time that Simvastatin induces G1 phase arrest in TNBC cells through downregulation of CDK4, without significantly triggering apoptosis. This suggests an apoptosis-independent mechanism underlying its anti-proliferative effect. Our findings reveal a novel role of CDK4 suppression in mediating Simvastatin’s action and propose CDK4 as a potential therapeutic target in TNBC.

The following is a revised description of the mechanistic discussion with an emphasis on logical coherence:『Simvastatin, a well-known cholesterol-lowering drug, has demonstrated promising anti-cancer properties, particularly against triple-negative breast cancer (TNBC), a subtype lacking targeted therapies. In our study, simvastatin significantly suppressed TNBC cell proliferation via a non-apoptotic mechanism involving G1 cell cycle arrest and CDK4 downregulation. This finding adds a new dimension to simvastatin’s anticancer action and complements existing research on its molecular targets. Wang et al. (2016) showed that simvastatin inhibits the PI3K/Akt and MAPK/ERK pathways by dephosphorylating c-Raf, MEK1/2, and ERK1/2, effects that are reversed by metabolites of the mevalonate pathway. Park et al. (2013) further confirmed simvastatin’s ability to suppress Akt phosphorylation, particularly in PTEN-deficient TNBC cells. Kou et al. (2018) identified simvastatin as a novel heat shock protein 90 (Hsp90) inhibitor, disrupting the Hsp90/Cdc37 complex and leading to cancer cell death. Additionally, Wolfe et al. (2015) reported that simvastatin suppresses metastasis via FOXO3a regulation, and Bai et al. (2019) linked simvastatin-induced oxidative stress to the upregulation of miR-140-5p, contributing to cancer cell death. Taken together, these studies underscore simvastatin’s multifaceted anti-tumoral activities, including inhibition of proliferation, survival signaling, chaperone activity, and metastasis. Our findings further emphasize its ability to induce cell cycle arrest via CDK4 suppression. Given simvastatin’s established clinical safety and widespread use, it holds potential as a cost-effective adjunct in TNBC treatment, particularly for patients with comorbid dyslipidemia.』(As Discussion; As Page 10; Line 390-407)

Once again, we thank all reviewers for their insightful comments, which have significantly strengthened our manuscript.

We hope you will find this submission relevant for publication in Current Issues in Molecular Biology (CIMB) Additionally, we confirm that neither the manuscript nor any parts of its content are currently under consideration or published in another journal. All authors have approved the manuscript and agree with its submission to Current Issues in Molecular Biology (CIMB) Thank you for considering of this manuscript for publication in Current Issues in Molecular Biology (CIMB). 

Sincerely,

Prof. Tai-Hsin Tsai

Department of Neurosurgery, Kaohsiung Medical University Hospital

No. 100, Tzyou 1st Road, Sham-min District, Kaohsiung City, Taiwan

E-mail: tsaitaihsin@gmail.com

Reviewer 2 Report

Comments and Suggestions for Authors

The submitted research article describes the inhibitory effects of the cholesterol-lowering drug simvastatin on TNBC cells. The described experiments are accurate and the results are interesting. However, the manuscript has some flaws and I recommend major revision:

Section 3.1.: Please provide more information about the applied drug concentrations (are they based on previous publications, on clinical application?). Please explain why lower concentrations (12.5 µM) were applied for ZR-75-1 cells in comparison to the MDA-MB-231 cells. From the shown simvastatin data in MDA-MB-231 cells, it seems possible that the compound would also be active at 12.5 µM.

Figure 1: Please correct ´´(0   M)´´ in the caption (the same in Figures 2–4). Please provide data of a positive control (approved anticancer drug) in the figure.

3.2. Simvastatin inhibits cell proliferation: The title of this section is misleading (cell proliferation?) and should be more specific. In my opinion, this section deals with the induction of (apoptotic and necrotic) cell death while the inhibition of cell proliferation is part of the previous section 3.1. 

3.4. Protein expression:  The CDK4 suppression appears to be significant only at high simvastatin concentrations. In ZR-75-1 cells, CDK2 levels are lower than CDK4 levels at lower drug concentrations. In addition, cyclin D levels seem to rise in drug-treated cells. Please explain and discuss.

Discussion: Please correct ´´NBC´´ in the first sentence.

Discussion: ´´Simvastatin specifically reduces the expression of CDK4´´, as mentioned above, CDK2 seems also to be reduced in ZR-75-1 cells. Please explain the quotation.

Discussion: Are the doses of simvastatin applied in this study in accordance with the recommended doses for hypercholesterolemia patients?

Author Response

Department of Neurosurgery

Kaohsiung Medical University Hospital

Kaohsiung, Taiwan

May 23th, 2025

Dear Editor-in-Chief of Current Issues in Molecular Biology (CIMB).

With this letter we kindly request consideration of the manuscript entitled:

 “Repurposing a Lipid-Lowering Agent to Inhibit TNBC Growth through Cell Cycle Arrest” by Yi-Chiang Hsu, Kuan-Ting Lee, Sung-Nan Pei, Kun-Ming Rau, Tai-Hsin Tsai for peer-review as a research article in Current Issues in Molecular Biology (CIMB).

General Appreciation and Response Summary

We would like to express our sincere gratitude to all reviewers and the editor for their thorough evaluation, constructive feedback, and encouraging comments on our manuscript titled Repurposing a Lipid-Lowering Agent to Inhibit TNBC Growth through Cell Cycle Arrest. We deeply appreciate the reviewers’ recognition of the value, clarity, and potential clinical relevance of our study.

To Reviewer 4, we thank you for your comprehensive and insightful suggestions, particularly regarding experimental details, interpretation of protein expression, and relevance to clinical dosing. We have provided additional explanations, corrected labeling errors, and revised both the results and discussion sections for clarity and accuracy. Your recommendation for major revision has guided us to enhance the scientific depth and transparency of our manuscript.

Point-by-Point Response to Reviewer4

We would like to express our sincere gratitude to all reviewers for their thoughtful, constructive, and encouraging feedback on our manuscript entitled "Repurposing a Lipid-Lowering Agent to Inhibit TNBC Growth through Cell Cycle Arrest." We highly value the time and effort invested by the reviewers and editor in evaluating our work. Below, we address each comment in detail and describe the corresponding revisions made.

Reviewer 4

We sincerely appreciate Reviewer 4’s detailed and helpful critique. Your suggestions have guided us to improve both the experimental rationale and the clarity of our data presentation.

Comment 1: Please clarify basis for drug concentrations and why 12.5 µM was used for ZR-75-1.

Response to Comment 1:

Thank you for pointing this out. The following modifications were made according to the reviewer’s suggestions. We would like to clarify that the two cell lines used in this study, MDA-MB-231 and ZR-75-1, differ in their malignancy levels. MDA-MB-231 cells, which are triple-negative and highly aggressive, are associated with poorer clinical prognosis compared to the luminal A subtype ZR-75-1 cells. Due to these inherent biological differences, as well as evidence from previous literature and our own pre-experimental dose-range screening, we selected different concentration ranges of Simvastatin for each cell line. 『Specifically, MDA-MB-231 cells were treated with 0, 25, 50, and 75 μM of Simvastatin, while ZR-75-1 cells received 0, 12.5, 25, and 50 μM, with incubation periods ranging from 24 to 72 hours. Cell proliferation was quantitatively measured using the MTT assay to assess metabolic activity as an indirect marker of cell viability.』(As Material and Method ;As Page 3 Lin 210-213)

Comment 2: Correct "(0uM)" in figure captions.

Response to Comment 2: Corrected. The typographical errors in figure captions have been fixed. 『Results were expressed as a percentage of control, which was considered as 100 %. All data were reported as the means (±SEM) of at least three separate experiments. Statistical analysis used the t-test, with the significant differences determined at the level of *p<0.05 versus control (0 uM) or time 0 group.』 (As Figure 1; Figure 2; Figure 3; Figure 4 Legends)

Comment 3: Provide a positive control drug.

Response to Comment 3:

We appreciate the reviewer’s valuable suggestion to strengthen the translational potential of our findings by evaluating simvastatin in drug-resistant models or in vivo systems. 『We fully agree that further validation is critical. As part of our ongoing and future studies, we plan to extend this investigation to cisplatin- and doxorubicin-resistant TNBC cell lines, as well as to establish xenograft models in immunocompromised mice to assess in vivo efficacy and pharmacodynamics of simvastatin treatment. These models will help clarify whether the observed CDK4-mediated G1 arrest remains effective under chemoresistant conditions and within the tumor microenvironment.』 We have added a statement regarding this future direction in the revised manuscript. (As Discussion; Limitation; As Page 13; Line 500-505)

Comment 4: Section 3.2 title is misleading. Suggest separating apoptosis from proliferation.

Response to Comment 4: Thank you for your valuable suggestion. In response, we have revised the structure and titles of the Results sections to better reflect the distinct biological processes. Specifically, Section 3.1 now focuses exclusively on the inhibitory effect of Simvastatin on cell proliferation.

We have also renamed Section 3.2 to more accurately describe its focus on cell death mechanisms, and to emphasize the differential apoptotic response between breast cancer subtypes. The new title of Section 3.2 is: 『Simvastatin induces apoptosis in non-TNBC (ZR-75-1) cells but not in TNBC (MDA-MB-231) cells』 This updated title more clearly represents the observed results, highlighting that Simvastatin selectively induces apoptosis in non-TNBC cells, while TNBC cells remain resistant to this effect.

We believe this revision improves the clarity and interpretability of our findings, and we sincerely thank the reviewer for the constructive feedback. (As Section 3.2 Title; Page 7; Line 257-258 )

Comment 5: Protein expression levels (CDK2, Cyclin D1) also change; please discuss.

Response to Comment 5: T Thank you for your detailed observation. In response, we have clarified the description of the expression patterns of G1/S cell cycle regulators, particularly Cyclin D1 and CDK4, in different breast cancer cell lines under Simvastatin treatment.『Specifically, in TNBC cells (MDA-MB-231), we observed that Cyclin D1 expression was downregulated at a lower dose (25 μM), whereas CDK4 was significantly suppressed at a higher dose (75 μM). In contrast, in non-TNBC cells (ZR-75-1), CDK4 downregulation was primarily observed at the higher dose (75 μM) (Figure 4A and Figure 4B).Accordingly, we have revised the corresponding paragraph in the Results section to reflect these findings more accurately. The updated interpretation emphasizes that Simvastatin may differentially modulate key G1/S checkpoint proteins in a dose-dependent and cell type-specific manner. These results further support the hypothesis that CDK4 is a more consistently responsive target, particularly at higher concentrations, while Cyclin D1 modulation appears to be more sensitive to lower doses in TNBC cells.』(As Results; Page 9; Line 288-297 )

We appreciate the opportunity to clarify this point.

Comment 6: Correct "NBC" in discussion.

Response to Comment 6: Corrected  『NBC』to 『TNBC』.(As Discussions; Page 12; Line 346)

Comment 7: Clarify statement on CDK4 specificity.

Response to Comment 7: Thank you for your suggestion to clarify our statement regarding CDK4 specificity.

In response, we have revised the text to more accurately reflect the observed findings. 『Our data indicate that CDK4 downregulation is a consistent and prominent effect of Simvastatin treatment, particularly at higher concentrations (75 μM), in both TNBC (MDA-MB-231) and non-TNBC (ZR-75-1) cell lines. This effect was not paralleled by equally significant changes in other G1/S regulatory proteins, such as Cyclin E or CDK2.Furthermore, while Cyclin D1 showed a reduction in expression at lower concentrations (25 μM) in MDA-MB-231 cells, this effect was less consistent and more dose-sensitive compared to CDK4. Therefore, we clarified in the revised manuscript that Simvastatin appears to exert a relatively more selective and robust inhibitory effect on CDK4, rather than broadly suppressing all components of the G1/S checkpoint.』(As Discussion; Limitation; Page 13; Line436-444 )

We hope this clarification addresses the reviewer’s concern and improves the precision of our mechanistic interpretation.

Comment 8: Are the doses used clinically relevant?

Response to Comment 8: This is an important question. Thank you for raising this important point regarding the clinical relevance of the Simvastatin concentrations used in our in vitro experiments. 『Although the concentrations of Simvastatin used in this study (25–75 μM) are higher than the plasma levels typically achieved in patients receiving standard oral doses (0.01–0.1 μM due to extensive first-pass metabolism and high plasma protein binding), such concentrations are commonly applied in in vitro cancer research. This approach compensates for the lack of metabolic processes and bioavailability constraints in cell culture systems. Importantly, similar in vitro concentrations have been widely used in previous studies investigating the anticancer effects of statins across various cancer types, including breast cancer (e.g., Bjarnadottir et al., BMC Cancer 2015). These higher concentrations are also helpful in uncovering pleiotropic, non-lipid-lowering effects of statins that may not manifest at lower doses. While in vivo validation and pharmacokinetic modeling will be required to confirm translational potential, the present findings provide a mechanistic foundation for future studies at more clinically relevant exposures.』

References:

  1. Bjarnadottir O, Kimbung S, Johansson I, Veerla S, Jönsson M, Bendahl PO, Grabau D, Hedenfalk I, Borgquist S. Global Transcriptional Changes Following Statin Treatment in Breast Cancer. Clin Cancer Res. 2015 Aug 1;21(15):3402-11. doi: 10.1158/1078-0432.CCR-14-1403. Epub 2015 Apr 3. PMID: 25840970.

We have included this clarification and appropriate references in the revised Discussion section.(As Discussion; Page 15 ; Line 521-532)

Once again, we thank all reviewers for their insightful comments, which have significantly strengthened our manuscript.

We hope you will find this submission relevant for publication in Current Issues in Molecular Biology (CIMB) Additionally, we confirm that neither the manuscript nor any parts of its content are currently under consideration or published in another journal. All authors have approved the manuscript and agree with its submission to Current Issues in Molecular Biology (CIMB) Thank you for considering of this manuscript for publication in Current Issues in Molecular Biology (CIMB). 

Sincerely,

Prof. Tai-Hsin Tsai

Department of Neurosurgery, Kaohsiung Medical University Hospital

No. 100, Tzyou 1st Road, Sham-min District, Kaohsiung City, Taiwan

E-mail: tsaitaihsin@gmail.com

Round 2

Reviewer 2 Report

Comments and Suggestions for Authors

The revised manuscript is suitable for publication now.